# Self Guided Exploration for Automatic and Diverse AI Supervision

## Abstract

Training large transformers using next-token prediction has given rise to groundbreaking advancements in AI. While this generative AI approach has produced impressive results, it heavily leans on human supervision. Even state-of-the-art AI models like ChatGPT depend on fine-tuning through human demonstrations, demanding extensive human input and domain expertise. This strong reliance on human oversight poses a significant hurdle to the advancement of AI innovation. To address this limitation, we propose a novel paradigm termed Exploratory AI (EAI) aimed at autonomously generating high-quality training data. Drawing inspiration from unsupervised reinforcement learning (RL) pretraining, EAI achieves exploration within the natural language space. We accomplish this by harnessing large language models to assess the novelty of generated content. Our approach employs two key components: an actor that generates novel content following exploration principles and a critic that evaluates the generated content, offering critiques to guide the actor. Empirical evaluations demonstrate that EAI significantly boosts model performance on complex reasoning tasks, addressing the limitations of human-intensive supervision.

## 1 Introduction

Training large transformers (Vaswani et al., 2017) using next token prediction has led to substantial AI advancements, as evidenced by the groundbreaking results they have produced (Schulman et al., 2022; OpenAI, 2023). While this generative AI approach has yielded remarkable AI results, it heavily relies on human supervision. For instance, state-of-the-art AI models including ChatGPT (Schulman et al., 2022) along with a range of other models (Chiang et al., 2023; Geng et al., 2023; Conover et al., 2023, *inter alia*), rely on fine-tuning through human demonstrations, demanding significant human involvement and domain expertise. This reliance on extensive human supervision presents a substantial challenge since human supervision requires domain expertise, is time consuming, and is tedious. Moreover, humans can struggle to provide reliable supervision in highly specialized domains. For instance, ChatGPT possesses a greater depth of knowledge than the average human, which makes it difficult to rely on humans to provide supervision for ChatGPT. Moreover, while our most advanced AI systems have made significant strides, they still necessitate thorough, human-guided processes to enhance their ability to answer factual or mathematical queries (Lightman et al., 2023). Yet, when it comes to more intricate and mission-critical tasks, such as navigating complex tax or law regulations, these challenges will demand even more specialized expertise and effort.

Prior works attempt to explore alternatives to human supervision, by using AI supervision instead. For example in mathematical reasoning, these studies propose sampling self generated solutions for human curated questions from large language models and employ techniques like rejection sampling, along with other techniques, to curate training data for the model (Cobbe et al., 2021; Ni et al., 2022; Bai et al., 2022; Huang et al., 2022; Zelikman et al., 2022; Yuan et al., 2023a, *inter alia*). While learning from such sampled content proves effective, a significant challenge persists: the sampled contents often lack the necessary diversity, resulting in a rapid saturation of the learning process (Yuan et al., 2023a; Zelikman et al., 2022). Moreover, the sampling approach has been confined to solutions exclusively, relying on human-curated questions, thus imposing constraints on the diversity of generated data.

To tackle these limitations, we propose a novel approach for using AI models to autonomously generate *diverse* data for learning purposes. This concept draws inspiration the APT algorithm (Liu and Abbeel, 2021b) designed for unsupervised RL pretraining (Singh et al., 2010; Laskin et al., 2021; Pathak et al., 2017). RL pretraining studies exploring in a reward-free environment to develop

Figure 1: Left and middle: Test accuracy on mathematical reasoning benchmark GSM8K. Baselines include Vicuna, supervised finetuning Vicuna on training set (denoted as SFT), and supervised finetuning Vicuna on rejection sampled model generated diverse solutions on training set (denoted as RFT). Our Exploratory AI (EAI) substantially outperforms all baselines. Right: Our approach EAI generates diverse data for learning by exploring with the guidance of principles and critiques.

skills for quickly maximize various downstream rewards. APT allows training RL agent to learn skills by autonomously explore reward free environment based on evaluating novelty of encountered states using particle based entropy estimation (Beirlant, 1997; Singh et al., 2003). Adapting APT to large language models presents several challenges, including computational complexity and the difficulty of learning reward functions and exploration policies (Gao et al., 2023; Cobbe et al., 2021). Rather than relying on traditional RL techniques, we harness the unique capabilities of large language models, such as their ability to learn from context and follow instructions. In essence, we use them to perform the roles of both a reward function and an exploration policy. Our approach, which we term Exploratory AI (EAI), involves two key components: an actor and a critic. The actor is responsible for generating novel content in natural language, while the critic evaluates this generated content and provides critiques to guide the actor's exploration. By evaluating the novelty of the generated contents, our method allows for effective exploration in the rich space of natural language. EAI can generate diverse data independently of human intervention. This makes it more scalable and automated, positioning it as a preferable alternative to methods like supervised finetuning or rejection sampling that depend on data curated by humans. Furthermore, EAI provides an interpretable window into the behavior and knowledge of the model. It sheds light on how well the model possesses knowledge and its reasoning behind generating novel questions. One can look at generations and their corresponding evaluations which provide valuable insights about how model generates and evaluates.

We evaluate our approach on mathematical reasoning benchmarks GSM8K (Cobbe et al., 2021) and MATH (Hendrycks et al., 2021), EAI substantially improves performance on challenging reasoning tasks, outperforming both human supervision and AI supervision baselines. In contrast to human supervision, our approach is autonomous and more scalable. When compared to prior state-of-the-art AI supervision baselines including RFT (Yuan et al., 2023a) and WizardMath (Luo et al., 2023), our method provides a straightforward yet highly effective solution for the generation of high-quality and diverse data.

Our contributions are two-fold: (a) In contrast to the predominant reliance on human supervision, our novel approach, EAI, leverages the capabilities of large language models to autonomously generate diverse high-quality training data. It achieves this by harnessing these models for self-guided exploration, inspired by unsupervised reinforcement learning pretraining. (b) We conduct an extensive series of experiments to systematically assess the effectiveness of EAI. Our results show that EAI substantially outperform prior human supervision and AI supervision state-of-the-arts, and significantly improve model performance.

## 2 EXPLORATORY AI FOR DIVERSE AI SUPERVISION

We present our approach for harnessing AI models to create AI supervision, in order to address the reliance on extensive human supervision.

Our method employs a dynamic interplay between generation and evaluation. This concept draws inspiration from unsupervised RL pretraining (URL) (Laskin et al., 2021) and particularly the APT algorithm (Liu and Abbeel, 2021b). RL pretraining studies exploring in a reward-free environment to develop skills for quickly maximizing various downstream rewards. APT allows training RL agent to learn skills by autonomously exploring a reward free environment based on evaluating novelty of encountered states using particle based entropy estimation (Beirlant, 1997; Singh et al., 2003).

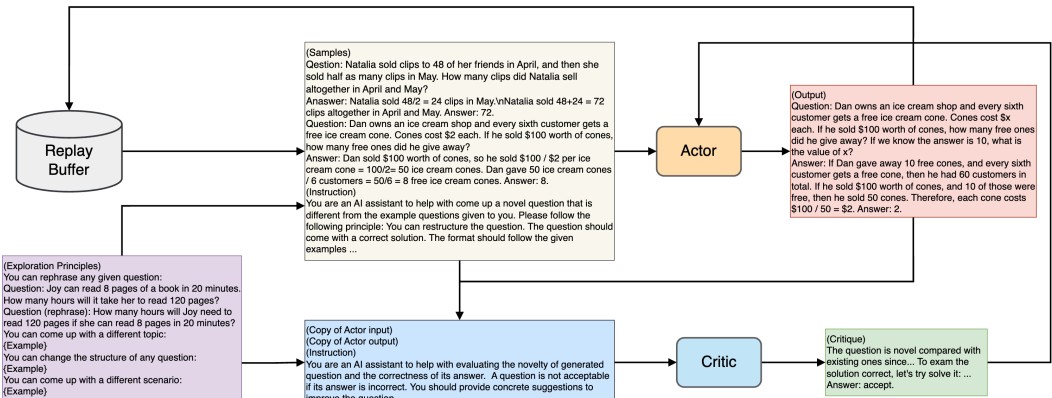

Figure 2: Generating diverse data in the Exploratory AI Framework. In the diagram, we demonstrate how the actor generates diverse content by conditioning on samples from the replay buffer and exploration principles. These principles include rephrasing question, coming up a novel topic, restructuring question, and coming up a new scenario, we provide examples associated with the principles to guide exploration. The actor's input and its generated output undergo evaluation by the critic. The critic assesses the novelty of the generated data; when the evaluation is favorable, the data is stored in the replay buffer. In cases where the evaluation does not meet the criteria, the critic provides critiques to guide the actor. The replay buffer can be initialized with a pre-existing human-created dataset (*e.g.*, GSM8K training set) or can remain empty for starting from scratch with zero-shot exploration.

Adapting APT directly to large language models presents several challenges, including grappling with computational complexity and the difficulty of learning reward functions and exploration policies (Gao et al., 2023; Cobbe et al., 2021). In response, we propose Exploratory AI (EAI), a novel approach that circumvents the need for direct reinforcement learning (RL) by harnessing the power of large language models for exploration. Our method explore the natural language space by employing these language models to assess the novelty of generated content and guide the exploration process. Our approach consists of two key components: an "actor" responsible for generating novel content and a "critic" that evaluates the actor's outputs and provides feedback to guide further content generation.

Concretely, we instruct the actor to generate content that diverges from samples from the replay buffer. The replay buffer can be initialized with a pre-existing human-created dataset (e.g., GSM8K training set) or can remain empty for zero-shot exploration. Similar to APT, we found having pre-existing samples accelerates learning and encourages the actor to have more long term exploratory behaviors. We then instruct the critic to assess the actor's outputs and provides critiques. This feedback loop guides the actor in refining and enhancing its content. This iterative process continues until it reaches a predefined maximum number of iterations, and the resulting outputs are stored in a dataset. The data can then be used for finetuning AI models.

**Actor prompt**

You are an AI assistant to help with come up a novel question that is different from the example questions given to you. The question should come with a correct solution. Please follow the given principle in generating the question. **{principle}**

**Critic prompt**

You are an AI assistant to help with evaluating the novelty of generated question and the correctness of its answer. A question is not acceptable if its answer is incorrect. You should provide concrete suggestions to improve the question. Explain your reasoning step by step and output final evaluation on novelty and correctness at the end. Follow the given principle on evaluating the novelty. **{principle}**

We equip both the actor and critic with a curated set of guiding principles to facilitate the generation and evaluation of diverse questions. These principles include rephrasing question, coming up a novel

topic, restructuring question, and coming up a new scenario, we provide examples associated with the principles to guide exploration.

---

**Principles for exploration**

**You can rephrase any given question:**
Question: Joy can read 8 pages of a book in 20 minutes. How many hours will it take her to read 120 pages?
Question (rephrase): How many hours will Joy need to read 120 pages if she can read 8 pages in 20 minutes?
**You can come up with a different topic:**
Question: Jack is stranded on a desert island. He wants some salt to season his fish. He collects 2 liters of seawater in an old bucket. If the water is 20% salt, how many ml of salt will Jack get when all the water evaporates?
Question (topic): Samantha is designing a circular garden in her backyard. The garden has a diameter of 8 meters. She wants to build a path around the garden that is 1 meter wide. What is the area of the path, in square meters, that Samantha will need to pave with stones or concrete?
**You can change the structure of any question:**
Question: Dan owns an ice cream shop and every sixth customer gets a free ice cream cone. Cones cost $2 each. If he sold $100 worth of cones, how many free ones did he give away?
Question (restructured): Dan owns an ice cream shop and every sixth customer gets a free ice cream cone. Cones cost $x each. If he sold $100 worth of cones, how many free ones did he give away? If we know the answer is 10, what is the value of x?
**You can come up with a different scenario:**
Question: Ed has 2 dogs, 3 cats and twice as many fish as cats and dogs combined. How many pets does Ed have in total?
Question (scenario): Sarah owns 4 bicycles, 2 skateboards, and three times as many pairs of rollerblades as bicycles and skateboards combined. How many wheeled sports equipment items does Sarah have in total?

---

While it's theoretically possible to provide all these principles to the model, in this study, we opt to a more controlled approach. To balance the quantity of generated data for each principle, we uniformly sample one principle at a time and input it to both the actor and critic. The actor is instructed to follow the principle (*e.g.*, restructuring the question) during question generation. Similarly, the critic's role is to evaluate the diversity, considering the selected principle. It's worth noting that the critic's principle is worded slightly differently from the exploration principle; for a detailed list, please refer to Appendix B. Our method is shown in Figure 2 and the algorithm is shown in Algorithm 1.

Exploratory AI has several attractive properties as an approach for facilitating AI supervision in language models:

1. EAI can generate diverse AI supervision for learning, independently of human input, making it more scalable compared with supervised finetuning or rejection sampling based on human curated data.

2. EAI provides an interpretable window into the behavior and knowledge of the model. It sheds light on how well the model possesses knowledge and its reasoning behind generating novel questions. One can look at generations and their corresponding evaluations which provide valuable insights about how model generates and evaluates.

3. EAI's versatility allows for a fusion of the best aspects of supervised finetuning and prompting. Users can prompt the model to focus on certain topics or aspects by directing actor and critic with different prompting strategies.

4. EAI demonstrates its effectiveness by excelling in mathematical reasoning tasks, as we will demonstrate in our experiments. Moreover, its capabilities are not limited to mathematics; it holds promise for a broad spectrum of language-related tasks in principle.

In empirical experiments, we will evaluate the utility of EAI for mathematical reasoning and analysis its effectiveness.

## 3 SETTING

We evaluate our method on the mathematical reasoning tasks, and achieve better results that EAI largely improve results and significantly outperforms prior state-of-the-arts.

**Benchmarks.** We evaluate our method on the mathematical reasoning tasks GSM8K and MATH. GSM8K exams model's mathematical reasoning capabilities, we finetune model on the training split,

---

**Algorithm 1** Exploratory AI for diverse AI supervision.

---

**Required:** One (or two) large language models $M$ for actor and critic.
Replay Buffer $B$, empty or optionally initialized with pre-existing data.
Initialize
**for** $i = 1$ **to** max iterations **do**
    Randomly sample data points from $B$
    Use preassigned principle or sample one principle.
    **for** $i = j$ **to** max rounds **do**
        Prompt the actor with the principle to generate content (a question and its answer) that in the same domain but diverge from the sampled inputs (questions and answers) sampled from $B$
        Prompt the critic with the principle to evaluate the diversity of generated question and correctness of answer, and decide whether to accept
        **if** Accepted **then**
            Save generated question and answer to $B$
            **break**
        **else**
            Continue to prompt actor with the critique as additional input
        **end if**
    **end for**
**end for**

---

and evaluate model on the test split. The GSM8k dataset includes around 7,500 training and 1,319 test math problems for high school-level students, involving basic arithmetic operations. Problems typically require 2 to 8 steps for a solution. The MATH dataset comprises 7,500 training and 5,000 challenging test problems from prestigious math competitions (AMC 10, AMC 12, AIME) covering various academic domains, including prealgebra, algebra, number theory, counting and probability, geometry, intermediate algebra, and precalculus.

**Baselines.** We compare our approach with (a) Base model including Vicuna 7B, 13B, and 30B (Chiang et al., 2023). Vicuna is LLaMA2 finetuned on user conversations shared online (ShareGPT). We use Vicuna as base model for all baselines and our method; (b) Supervised finetuning (SFT) on training set of the original GSM8K or MATH, in which a language model is finetuned on human written exemplars of questions–answers pairs. SFT has been widely used in prior works for improving language models mathematical reasoning (Lewkowycz et al., 2022; Touvron et al., 2023; OpenAI, 2023, *inter alia*) and following user intention (Geng et al., 2023; Conover et al., 2023, *inter alia*). We also compare with WizardMath (Luo et al., 2023) which does SFT on ChatGPT annotated questions and solutions, as well as MAmmoTH (Yue et al., 2023) which uses GPT4 annotated solutions; (c) Rejection sampling finetuning (RFT) (Yuan et al., 2023a) which applies supervised finetuning on rejection sampled model generated data. We provide baseline scores for SFT and RFT from both their original papers and our implementations using Vicuna, ensuring a fair and comprehensive comparison; (d) Proprietary models including GPT-4 (OpenAI, 2023), ChatGPT (Schulman et al., 2022), and Claude2 (Anthropic, 2023). All baselines are evaluated by prompting them to output step by step reasoning followed by final answers (Wei et al., 2022).

**Generated data size.** We sample roughly the same amount of data for each principle outlined in Section 2. To optimize computational cost, we have set the number of interaction rounds in Algorithm 1 to a maximum of two. Our preliminary experiments revealed that this two-round interaction is typically sufficient for the actor to produce high-quality and diverse data. For each of the four principles – 'rephrase question', 'introduce a new topic', 'restructure the question', and 'introduce a new scenario' – we generate approximately 25,000 samples for GSM8K and approximately 15,000 samples for MATH. The generation on 8 A100 80GB GPUs take from 40 to 200 hours depending on the model size and the specific principles applied.

## 4 RESULTS

**Benchmarks on math reasoning.** The results presented in the Table 1 demonstrate the notable effectiveness of the method EAI in the context of pass@1 performance on the GSM8k and MATH datasets. A key highlight is the absolute improvement of EAI over previous state-of-the-art methods,

Table 1: Results of pass@1 (%) on GSM8k and MATH. In this study, to ensure equitable and cohesive evaluations, we report the scores of all models under the same settings of greedy decoding. Bold numbers in red are the absolute improvement of EAI over prior state-of-the-arts. Notably, EAI outperforms state-of-the-arts both when using ChatGPT for exploration to supervise LLaMA2 and when using Vicuna to supervise itself.

| Finetune Model | Method | AI supervision | Data amount | Params | GSM8K | MATH |
|---|---|---|---|---|---|---|
| - | GPT-4 | - | - | - | 92.0 | 42.5 |
| | ChatGPT | - | - | - | 80.8 | 34.1 |
| | Claude 2 | - | - | - | 88.0 | 32.5 |
| LLaMA2 | LLaMA2 | - | - | 7B | 14.6 | 2.5 |
| | | | | 13B | 28.7 | 3.9 |
| | SFT | - | 7.5K | 7B | 41.6 | - |
| | | | | 13B | 50.0 | - |
| | RFT | LLaMA2 | 47K | 7B | 47.5 | 5.6 |
| | | | | 13B | 54.8 | 9.6 |
| | WizardMath | ChatGPT | 96K | 7B | 54.9 | 10.7 |
| | | | | 13B | 63.9 | 14.0 |
| | MAmmoTH | GPT4 | 260K | 7B | 51.7 | 31.2 |
| | | | | 13B | 61.7 | 36.0 |
| | **EAI** | GPT4 | 96K | 7B | **56.6** (+1.7) | **11.6** (+0.9) |
| | | | | 13B | **65.2** (+1.3) | **15.1** (+1.1) |
| Vicuna | Vicuna | - | - | 7B | 24.4 | 2.6 |
| | | | | 13B | 39.8 | 5.8 |
| | SFT | - | 7.5K | 7B | 42.0 | 4.6 |
| | | | | 13B | 50.8 | 7.9 |
| | RFT | Vicuna | 48K | 7B | 48.1 | 5.9 |
| | | | | 13B | 56.3 | 9.3 |
| | **EAI** | Vicuna | 48K | 7B | **52.9** (+4.8) | **8.6** (+2.7) |
| | | | | 13B | **60.5** (+4.2) | **11.4** (+2.1) |

emphasized in bold red numbers. Specifically, EAI exhibits superior performance in two distinct scenarios: firstly, when ChatGPT is used to supervise LLaMA2, and secondly, when Vicuna supervises itself. The table provides a comprehensive comparison across various models and methods, including GPT-4, ChatGPT, Claude 2, LLaMA2 (with and without SFT and RFT), and Vicuna. The performance of EAI, especially in the LLaMA2 and Vicuna settings, shows marked improvements in both the GSM8K and MATH datasets. For instance, in the LLaMA2 model, EAI achieves a significant gain in both datasets, irrespective of the number of parameters (7B or 13B). This trend is consistent in the Vicuna model as well, where EAI again shows superior performance compared to the baseline Vicuna model. These results underscore the efficacy of EAI in leveraging AI supervision to enhance model performance. The gains are prominent across model sizes, indicating EAI's scalability and effectiveness in handling complex AI models. This result on the GSM8k and MATH datasets provides

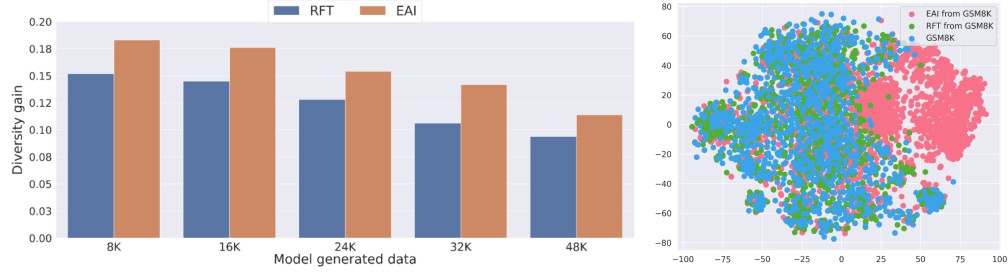

Figure 3: (Left): Comparison of diversity gains achieved by adding generated data to the GSM8K training set. EAI outperforms other baselines in terms of diversity. (Right): t-SNE comparison of human-curated GSM8K, RFT, and EAI-generated outputs, depicting embeddings of questions.

compelling evidence of the effectiveness of generating diverse AI supervision by EAI to enhance complex problem-solving tasks.

**Comparison of diversity.** We evaluate EAI in terms of the diversity of generated data. We compare RFT and EAI in terms of the submodularity diversity gain (Bilmes, 2022; Murphy, 2023). This metric serves as an indicator of the extent to which the generated data contribute to the overall diversity of the dataset. A higher diversity gain suggests that the newly generated questions exhibit greater dissimilarity from the existing dataset. We measure the gain over GSM8K training set by varying the amount of generated content. We use OpenAI GPT embedding `text-embedding-ada-002` to encode the data. The results of diversity gain and t-SNE are depicted in Figure 3 demonstrate that EAI consistently outperforms RFT in terms of diversity, thereby providing a more diverse set of generated data.

**Effect of sampled inputs.** The Table 2 presents the results of an experiment examining the impact of varying the number of samples on GSM8K and MATH. As the number of samples increases from 0 to 8, we observe a steady incremental improvement on both GSM8K and MATH. On GSM8K, the performance rises from 50.1 to 52.9. On MATH, the effect is more pronounced. These results suggest that increasing the number of samples has a positive effect on both GSM8K and MATH, highlighting the significance of larger sample size for in context exploration.

Table 2: Effect of different number of samples from replay buffer.

| Number | 0 | 1 | 4 | 8 |
|---|---|---|---|---|
| GSM8K | 50.1 | 50.8 | 51.9 | **52.9** |
| MATH | 6.6 | 7.1 | 7.5 | **8.6** |

**Scaling with generated data.** We assess the performance of EAI in terms of sample efficiency on the GSM8K dataset. Our primary focus lies in understanding how the results evolve in response to varying amounts of generated data. Sample efficiency holds paramount importance, given that autoregressive data generation is inefficient. Enhanced sample efficiency broadens the practical utility of our approach in real-world applications. The results depicted in Figure 4 clearly illustrate a significant advantage for EAI over the previous state-of-the-art RFT. Notably, as more data is employed, RFT exhibits improved performance, but its sample efficiency lags behind EAI by a substantial margin. At just 16K data points, EAI outperforms RFT's performance at 48K data points, achieving more than a 3x higher level of sample efficiency.

**Evaluating the effect of exploration principles.** The results of varying exploration principles, as shown in Table 3, reveal some interesting insights. When all principles are in place (✓ for rephrase, new topic, restructure, and new scenario), the model performs at its best on GSM8K and MATH. This suggests that using all principles simultaneously leads to the most favorable outcomes. Among the principles, the most critical ones appear to be "rephrase" and "restructure", as seen when one of them is removed (✗). Without "rephrase" the performance drops on both datasets, emphasizing that the ability to rephrase and generate diverse content is crucial. Similarly, the omission of "restructure" leads to a significant drop in MATH scores, highlighting the significance of introducing novel question-structuring approaches for solving more challenging problems.

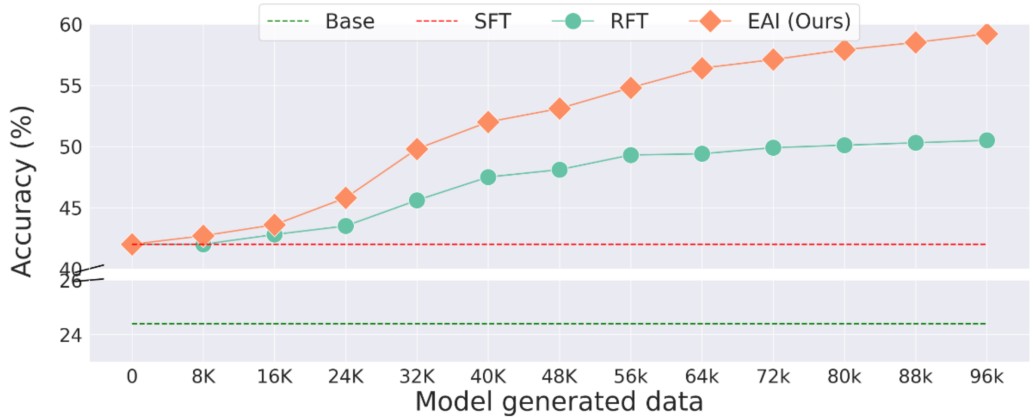

Figure 4: Data scaling on GSM8K. Shown are GSM8K accuracy with different amount of generated data. EAI generates high quality data for learning and scales well with data.

Table 3: Effect of different exploration principles on GSM8K and MATH.

| rephrase | new topic | restructure | new scenario | GSM8K | MATH |
|:---:|:---:|:---:|:---:|:---:|:---:|
| ✓ | ✓ | ✓ | ✓ | **52.9** | **8.6** |
| ✗ | ✓ | ✓ | ✓ | 48.8 | 7.1 |
| ✓ | ✗ | ✓ | ✓ | 49.7 | 7.8 |
| ✓ | ✓ | ✗ | ✓ | 48.9 | 6.9 |
| ✓ | ✓ | ✓ | ✗ | 49.5 | 7.5 |
| ✓ | ✗ | ✗ | ✗ | 48.1 | 6.3 |
| ✗ | ✓ | ✗ | ✗ | 47.6 | 6.0 |
| ✗ | ✗ | ✓ | ✗ | 48.5 | 6.2 |
| ✗ | ✗ | ✗ | ✓ | 47.8 | 6.3 |

**Evaluating the effect of critic.** We remove the critic from the framework and evaluate its performance. To ensure a fair comparison, we follow the same procedure as with the default EAI and generate an equal amount of data. We evaluate on GSM8K, MATH, MBPP, and HumanEval. All methods are based on LLaMA2 (Touvron et al., 2023). We compare EAI and EAI w/o critic with Self-instruct (Wang et al., 2022) and RFT Yuan et al. (2023a). Self-instruct involves a loop of generation and filtering which resembles EAI's actor-critic, except that EAI comes with principles guided in context exploration. Comparing EAI w/o critic with Self-instruct shows the effectiveness of the principles guided in context exploration component in EAI framework. Comparing EAI with EAI w/o critic can show the importance of the critic. RFT relies on sampling-based exploration; comparing EAI w/o critic with it shows the effectiveness of in-context exploration and comparing EAI with it further reveals the importance of the critic. The results in Table 4 show that EAI w/o critic significantly outperforms all baselines, showing the effectiveness of principles guided in context exploration. Adding the critic back to the EAI framework further substantially improves the results, achieving significantly better results than Self-instruct, RFT, and EAI w/o critic. A qualitative analysis provided in Appendix A reveals how the critic aids in guiding exploration. Both quantitative and qualitative results show that the critic is important for achieving the best exploration; removing the critic leads to substantially lower results.

**Scaling with human annotation size.** Figure 5 illustrates the results obtained when utilizing varying amounts of human annotation data from the GSM8K training set. We employ three different approaches in our experiments: SFT which directly finetunes the base model, Vicuna-7B, on the provided data. RFT which leverages the provided data to perform rejection sampling from the model. EAI which utilizes the provided data to initialize a replay buffer and explore new content for training. The results consistently demonstrate that EAI significantly outperforms all the baseline methods across various levels of human annotation data, underscoring its efficacy in generating high-quality training data. Remarkably, our experiments reveal that EAI performs admirably even in the absence of any human annotations, hinting at the potential to entirely eliminate the need for human intervention in the process.

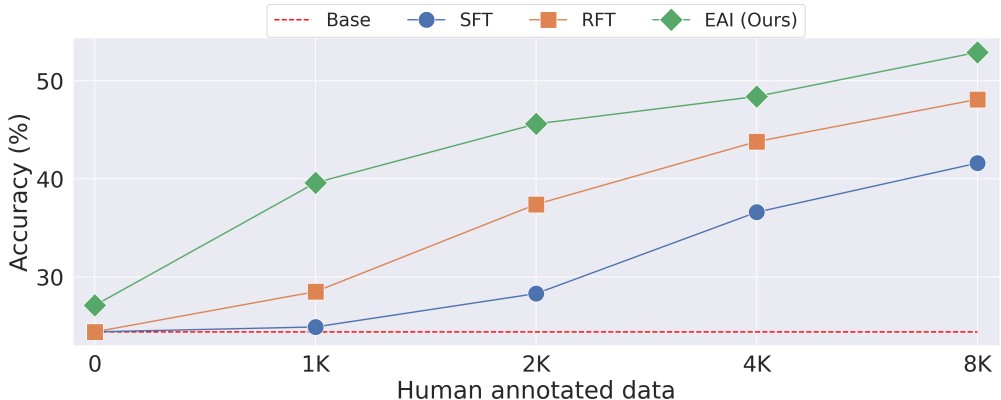

Figure 5: Performance on GSM8K with different amount of human annotated data. EAI performs well even without human annotation and scales well with more human provided annotations.

Table 4: Evaluation of the effectiveness of critic.

|  | LLaMA2 | Self-Instruct | RFT | EAI w/o critic | **EAI** |
|---|---|---|---|---|---|
| GSM8K (%) | 14.6 | 43.4 | 47.5 | 50.5 | **52.9** |
| MATH (%) | 2.5 | 3.9 | 5.6 | 7.1 | **8.6** |
| MBPP (%) | 26.1 | 33.7 | 41.8 | 42.5 | **44.6** |
| HumanEval (%) | 11.9 | 12.8 | 13.9 | 14.5 | **15.8** |

**Benchmark on code generation.** Having evaluated the effectiveness of EAI in improving mathematical reasoning, we further evaluate its application in a different domain: code generation. Unlike the math reasoning task, where the model generates the reasoning path and final answer, in code generation the model needs to output a Python program in response to a given question, such as 'Write a function to convert degrees to radians.' Specifically, we apply EAI to the MBPP task (Austin et al., 2021), following the same procedure as before. This task comprises around 1,000 crowd-sourced Python programming problems designed to be solvable by entry-level programmers. It covers programming fundamentals, standard library functionality, and more. Each problem includes a task description, a code solution, and three automated test cases. We collected 18,000 examples using EAI and RFT based on the MBPP training split, respectively. We then compared EAI with supervised finetuning as well as RFT on the training split. Subsequently, we evaluated the models on the MBPP test split and conducted zero-shot evaluation on HumanEval (Chen et al., 2021). For evaluation, we used LLaMA2 (Touvron et al., 2023), which is pretrained on text, and CodeLLaMA (Roziere et al., 2023), which is further pretrained on code, representing different model choices. Table 5 shows the results of 0-shot HumanEval and 3-shot MBPP. EAI substantially outperforms the baselines and significantly improves both LLaMA2 and CodeLLaMA. This confirms the effectiveness of EAI in exploration and in improving code generation performance.

Table 5: Evaluations on code generation tasks. LLaMA2 and CodeLLaMA are pretrained models. SFT, RFT, EAI are trained using MBPP training split. All methods are evaluated using MBPP test split, and HumanEval dataset. Red numbers are absolute increase compared with best performing baselines.

|  | LLaMA2 | LLaMA2+SFT | LLaMA2+RFT | LLaMA2+**EAI** |
|---|---|---|---|---|
| MBPP (%) | 26.1 | 38.5 | 41.8 | **44.6** (+2.8) |
| HumanEval | 11.9 | 13.2 | 13.9 | **16.2** (+2.3) |

|  | CodeLLaMA | CodeLLaMA+SFT | CodeLLaMA+RFT | CodeLLaMA+**EAI** |
|---|---|---|---|---|
| MBPP (%) | 52.5 | 54.3 | 55.6 | **58.9** (+3.3) |
| HumanEval | 31.1 | 33.5 | 36.1 | **39.5** (+3.4) |

## 5 RELATED WORK

Transformers (Vaswani et al., 2017) trained using next token prediction have gave rise to many state-of-the-art AI systems (Schulman et al., 2022; OpenAI, 2023). The remarkable AI results achieved with this generative AI approach heavily hinge upon the availability of diverse and high-quality data. For instance, state-of-the-art AI models including ChatGPT (Schulman et al., 2022) and GPT4 (OpenAI, 2023) along with a range of other open source models such as Vicuna, Koala, and Dolly (Conover et al., 2023; Geng et al., 2023; Chiang et al., 2023, *inter alia*), require extensive finetuning through human demonstrations. This process involves human conversations with ChatGPT or written demonstrations, demanding significant human involvement and domain expertise. Previous research has explored various avenues to enhance performance and sample efficiency, as well as alternative sources of supervision. To align with human preferences, there has been active research into developing simple algorithms for learning from human preferences (Liu et al., 2023; Yuan et al., 2023b; Dong et al., 2023; Touvron et al., 2023, *inter alia*). In contrast to human demonstrations or feedback, another line of work explores the utilization of environmental feedback, such as unit test errors (Le et al., 2022; Chen et al., 2023; Shinn et al., 2023), which has demonstrated improved results in coding tasks, or using using LLMs to provide AI supervision based exploration techniques for applications in solving games (Du et al., 2023; Wang et al., 2023) and demonstrate improved results. Furthermore, some prior research leveraged proprietary APIs to indirectly obtain high-quality human data, enhancing model capabilities in areas like instruction following (Wang et al., 2022; Xu et al.,

2023; Taori et al., 2023; Geng et al., 2023; Chiang et al., 2023) and mathematical reasoning (Luo et al., 2023; Mukherjee et al., 2023; Yue et al., 2023, *inter alia*). Another line of research explores the use of models to supervise themselves (Sun et al., 2023; Madaan et al., 2023; Huang et al., 2022; Bai et al., 2022; Yuan et al., 2023a), yielding improved results in reasoning tasks and alignment with human preferences. Our work focuses on generating diverse and high-quality data using AI models and we demonstrate applying our proposed approach to enhance open-source models by having them self-generate learning data. Our approach's exploration technique is related to unsupervised RL based exploration (Singh et al., 2010; Liu and Abbeel, 2021a;b; Campos et al., 2021; Pathak et al., 2017; Mutti et al., 2020; Eysenbach et al., 2018; Park et al., 2023; Rajeswar et al., 2023, *inter alia*), however, our method does not require training RL agent directly and relies on LLMs to explore. Additionally, some works have delved into more detailed forms of human supervision (Lightman et al., 2023), demonstrating that LLMs benefit more from step-by-step process-based supervision than sparse outcome-based supervision. Our research centers on the data dimension, with a specific emphasis on harnessing AI models to generate diverse high-quality AI supervision. To this end, we introduce an actor-critic based approach based with human provided principles for automating the exploration process.

## 6 CONCLUSION

In this work we propose an approach to automatically generate diverse, high-quality data from AI models. Our approach Exploratory AI consists of prompting an actor model to generate diverse contents that are different from existing contents, and using a critic model for evaluating the novelty of generated data and providing critiques to guide the exploration process. Experimental evaluations confirms the effectiveness of EAI, demonstrating its capacity to generate diverse content and substantially enhance model performance on GSM8K and MATH datasets.

In terms of future prospects, our approach of generating diverse content with AI models opens up interesting possibilities, such as extending EAI to evaluate novelty across the entire data buffer, employing either a brute force approach (evaluating all data through a critic) or employing embedding similarity search techniques. Moreover, there's potential in extending EAI to incorporate multiple actor and critic models, or in experimenting with various strategies to enhance exploration. It would also be interesting to apply our method to proprietary APIs to source even more diverse and higher-quality data.

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

# A  CASE STUDY OF EAI

We conduct a qualitative study how actor and critic accomplish exploration. On GSM8K, we manually inspect 100 examples randomly sampled from EAI explored dataset to better understand how does the exploration work. We observed that the critic excels at assessing the novelty of generated questions, providing precise feedback to enhance them. The actor effectively utilizes the critic's feedback to refine the generated questions. Of the 100 samples reviewed, in 24 instances, the critic initially found the questions lacking in novelty and provided suggestions for improvement. Out of these, 19 were subsequently refined and accepted after revisions. Below is a representative example of this interaction where the critic offers advice to significantly differentiate the generated question. Here, the critic recommends incorporating the mathematical concept of volumes and presenting an alternative scenario

```
[Critic guides exploration]
(First round)
```
Actor's generation: Question: Jake is planning to buy a new skateboard which costs $200. Jake only has a quarter of the money he requires. His brother decided to give him $20 for it, and his uncle three times as much as his brother. How much more does Jake need to raise to purchase the skateboard? (answer omitted)

Critic's critique: The new question you've provided is somewhat similar in structure to the existing questions, as it involves basic arithmetic operations (multiplication, division, addition, and subtraction). While it introduces a different context with hats and balls, the underlying math principles are similar to those used in the existing questions. To make the new question substantially different, you could consider introducing more complex mathematical concepts such as volumes or using a different scenario altogether. ✗
```
(Second round)
```
Actor's generation: An artist is creating a sculpture using various geometric shapes. The sculpture consists of a cube, a sphere, a cylinder, and a pyramid. The cube has a side length of 6 units, the sphere has a radius of 4 units, the cylinder has a height of 8 units and a base radius of 3 units, and the pyramid has a base edge length of 5 units and a height of 7 units. Calculate the total volume of the artist's sculpture. A: omitted

Critic's critique: The new question is substantially different from the existing questions. It involves a slightly more complex problem-solving process. It is good to be added to math questions. The decision is accept. ✓

We also conducted a study to determine how effective the critic is at evaluating the correctness of answers. Of the 100 critiques assessed, the critic accurately identified incorrectness in 13 actor generations and offered critiques to correct 10 of them. However, there were 7 instances where the critic failed to recognize the errors. The remaining actor generations were accurate. It's worth noting that, due to the uniform usage of the four exploration principles, out of the 100 actor generations, 50 were either paraphrased questions or questions with new scenarios, these 50 were less prone to errors as they didn't necessitate significant changes to the mathematical structure. In fact, only 2 solutions among them were incorrect. On the other hand, those that involved restructuring or tackled new topics were more error-prone: 7 from restructuring and 4 from new topics were incorrect. Intriguingly, the critic corrected 6 out of the 7 restructured ones, but only 1 out of the 4 new topics. This might suggest that questions on new topics are more open-ended and might need multiple rounds of interactions for accurate evaluation. Further refinement in this verification process will be the focus of our future work.

Our analysis revealed that the critic excels in evaluation rather than generation. This observation is exemplified in the quoted example, it can be seen that even when the model cannot directly answer the question, it can effectively assess the correctness of the actor's response. This discovery is intriguing, as the actor's response effectively serves as a 'hint'. This observation aligns with prior works that found models are better at evaluating answer (or improving an answer) than generating it from scratch in various settings (Chen et al., 2023; Madaan et al., 2023; Liu et al., 2023; Cobbe et al., 2021, *inter alia*).

```
[Critic cannot solve question directly]
```
Input: A magician has 5 hats, each with a different number of balls inside. The first hat has 3 balls, the second hat has twice as many balls as the first, the third hat has half as many balls as the second, the fourth hat has 5 balls, and the fifth hat has 4 times as many balls as the fourth. How many balls are in the magician's hats altogether?

`Output:` There are 3 + 2 * 3 = 7 balls in the first two hats. There are 5 - 2 = 3 balls in the third hat. There are 4 * 3 = 12 balls in the fourth and fifth hats. In total, there are 7 + 3 + 12 = 22 balls in the magician's hats. Answer: 22. ✗

`[Critic can evaluate actor's answer]`
`Input:` Please review whether the answer for the following question is correct. Q: A magician has 5 hats, each with a different number of balls inside. The first hat has 3 balls, the second hat has twice as many balls as the first, the third hat has half as many balls as the second, the fourth hat has 5 balls, and the fifth hat has 4 times as many balls as the fourth. How many balls are in the magician's hats altogether? A: There are 3 + 2 x 3/2 = 3 + 3 = 6 balls in the first two hats. There are 5 - 3/2 = 5 - 1.5 = 3.5 balls in the third hat. There are 4 x 5 = 20 balls in the fourth hat. In total, there are 6 + 3.5 + 20 = 29.5 balls. Since you can't have a fraction of a ball, you would need to round up to the nearest whole number of balls. Therefore, there are 29 balls in the magician's hats altogether.

`Output:` The magician has 5 hats, each with a different number of balls inside. The first hat has 3 balls, the second hat has twice as many balls as the first, so it has 3 * 2 = 6 balls. The third hat has half as many balls as the second, so it has 6 / 2 = 3 balls. The fourth hat has 5 balls. The fifth hat has 4 times as many balls as the fourth, so it has 5 * 4 = 20 balls. In total, there are 3 + 6 + 3 + 5 + 20 = 37 balls. Since you can't have a fraction of a ball, you would need to round up to the nearest whole number of balls. Therefore, there are 37 balls in the magician's hats altogether. The answer provided is incorrect. ✓

## B  PROMPT

The critic's principle is based on the exploration principle but is worded slightly differently from the actor's exploration principle, in order to tell critic what kind of new questions are diverse and acceptable. The critic will use the principle to evaluate diversity of questions and evaluate the correctness of answer.

---

**Principles for critique**

**A different topic is acceptable:**
Question: Jack is stranded on a desert island. He wants some salt to season his fish. He collects 2 liters of seawater in an old bucket. If the water is 20% salt, how many ml of salt will Jack get when all the water evaporates?
Question (topic): Samantha is designing a circular garden in her backyard. The garden has a diameter of 8 meters. She wants to build a path around the garden that is 1 meter wide. What is the area of the path, in square meters, that Samantha will need to pave with stones or concrete?
**A question with different structure is acceptable:**
*Question*: Dan owns an ice cream shop and every sixth customer gets a free ice cream cone. Cones cost $2 each. If he sold $100 worth of cones, how many free ones did he give away?
*Question (restructured)*: Dan owns an ice cream shop and every sixth customer gets a free ice cream cone. Cones cost $x each. If he sold $100 worth of cones, how many free ones did he give away? If we know the answer is 10, what is the value of x?
**Rephrased question is acceptable:**
Question: Joy can read 8 pages of a book in 20 minutes. How many hours will it take her to read 120 pages?
Question (rephrase): How many hours will Joy need to read 120 pages if she can read 8 pages in 20 minutes?
**A different scenario is acceptable:**
Question: Ed has 2 dogs, 3 cats and twice as many fish as cats and dogs combined. How many pets does Ed have in total?
Question (scenario): Sarah owns 4 bicycles, 2 skateboards, and three times as many pairs of rollerblades as bicycles and skateboards combined. How many wheeled sports equipment items does Sarah have in total?

---

## C  EXPERIMENT DETAILS

We use a temperature of 0.7 for the actor during exploration as in prior work (Cobbe et al., 2021), and we sample 10 actor generations for every batch of samples from the replay buffer. We use a temperature of 0.0 for the critic since we found that it performs best. Following prior work (Yuan et al., 2023a), we filter out reasoning paths with incorrect answers or calculations—based on Python evaluation—for the 'paraphrasing' and 'new scenarios' exploration categories. However, we do not apply this filter to the 'restructuring' or 'new topics' exploration categories, as we do not have ground truth answers for these two categories. The evaluations for all baselines and our approach are conducted with deterministic sampling following prior work and report maj1@1 (accuracy) across all experiments. We follow prior work by conducting evaluations using deterministic sampling for both

our approach and the baseline methods. We report maj1@1 accuracy across all experimental setups. All models are trained with the same hyperparameters: global batch size = 128, learning rate = 2e-5, epochs = 3, sequence length = 2048. The training is done with 8x A100 80GB GPUs.

