# OpenReview forum: "Self Guided Exploration for Automatic and Diverse AI Supervision"
_ICLR.cc/2024/Conference — Submitted to ICLR 2024_

### Official Review · Reviewer_1gMq · 2023-10-31

**Soundness:** 3 good
**Presentation:** 3 good
**Contribution:** 3 good
**Rating:** 6
**Confidence:** 3

**Summary:**

This paper proposes Exploratory AI (EAI), a novel approach for using large language models to autonomously generate diverse training data through self-guided exploration. EAI employs an actor-critic framework where the actor generates novel content and the critic evaluates it, providing feedback to guide further exploration. The method is inspired by unsupervised reinforcement learning pretraining (APT) and harnesses language models to assess the novelty of generated text. Empirical evaluations on mathematical reasoning datasets GSM8k and MATH demonstrate that EAI can produce high-quality and diverse data, leading to improved performance over both human-supervised and prior AI-supervised baselines. In general, EAI provides a simple yet effective paradigm for automated and diverse data generation without human involvement.

**Strengths:**

- The paper is well-motivated and easy to follow. To address the reliance of current large models on extensive human supervision and fine-tuning, the paper aims to generate high-quality training data automatically.
- The proposed actor-critic framework is simple yet effective. Experiments on mathematical reasoning benchmarks are impressive. EAI outperforms supervised fintuning (SFT) and rejection sampling finetuning by a large margin.
- The experiments are well conducted. While it is challenging to evalute the quality of generated data, the authors did some attempts to showcase the effectiveness of the proposed paradigm, including quantitative diversity measure and case studies. Moreover, the analysis on sample efficiency and scalability with human annotations helps verify the robustness of EAI paradigm.

**Weaknesses:**

- From Table 3, we can observe that "rephrase" and "restructure" play an more important role than the other two principles. This indicates the model does not see many variations of input data. Will some simple augmentations on the data improve the performance? The prompts for actor and critic encode the human priors, which is similar to encode those priors with specific rules. Some comparisons with human designed rules would be interesting.
Moreover, the ablation is not complete, how about only do one principle at a time? Will the performance drop a lot? Could we do some classfication on the generated data (xx% rephrase, xx% new scenario, or so)? It is also helpful to release the questions/answers generated on GSM8k and MATH for future comparisons and analysis.
- The paper only studies EAI on mathematical reasoning task although the proposed paradigm is quite simple and general. More thorough study on different tasks would better demonstrate the effectiveness of EAI in generating new data. It is also interesting that for other tasks, which types of exploration / critique strategies are required.
- Can we introduce the actor-critic paradigm during inference, will this procedural inference help the reasoning on GSM8k?

Others:
- The labels for Figure 4 are not correct. there are two SFTs.

**Questions:**

See weaknesses. In general, this paper proposes an interesting paradigm to generate data and achieve impressive results. More rigorious study on this paradigm to test its effectiveness and generability on other tasks/models are beneficial.

---

> ### Author Response · Authors · 2023-11-20
> **Response to reviewer 1gMq**
>
> Dear reviewer 1gMq,
>
>
> Thank you so much for your positive review and insightful comments. We appreciate your feedback and suggestions, which we have incorporated into the revised version to improve it.
>
>
> **#1: Can exploration principles be used for simple data augmentation to improve performance?**
>
> It's an insightful question whether data augmentation based on simple rules improves performance. Augmenting text with simple rules is an interesting baseline to consider; we have conducted experiments to study it. Unlike image augmentation, text augmentation is nontrivial; therefore, we opted to use Vicuna, which is the same model used in EAI, to augment the GSM8K and MATH training split. We augment each data point multiple times to obtain the same number of data points as generated by EAI, and use EAI's exploration principles as rules for data augmentation. Concretely, for each question, we ask Vicuna to follow a principle such as 'rephrasing' to augment the question, and we sample multiple responses. The results shown in the table below reveal that data augmentation improves performance, especially when all principles are combined, which outperforms human-curated data. It also shows that EAI performs substantially better than all the data augmentation variants, highlighting the importance of in-context exploration and the effectiveness of the EAI framework.
>
> |        | Vicuna | SFT  | Data Aug (restructure) | Data Aug (rephrase) | Data Aug (all) | EAI  |
> |--------|--------|------|------------------------|---------------------|----------------|------|
> | GSM8K  | 24.4   | 42.0 | 42.8                   | 42.5                | 43.4           | 52.9 |
> | MATH   | 2.6    | 4.6  | 5.2                    | 4.9                 | 5.5            | 8.6  |
>
>
>
> **#2: Some comparisons with human designed rules would be interesting. Moreover, the ablation is not complete, how about only do one principle at a time? Will the performance drop a lot?**
>
> Thank you for the suggestion to study one principle at a time and to classify the generated data into categories. To this end, we have conducted additional ablation experiments to study how each principle works. The results are included in Table 3, which is attached below. This analysis reveals that removing any one of the principles reduces performance; using only one of them is suboptimal; among the principles, rephrasing and restructuring are the most effective; and combining all principles achieves the best results.
>
>
> | rephrase | new topic | restructure | new scenario | GSM8K | MATH |
> |----------|-----------|--------------|--------------|-------|------|
> | ✓        | ✓         | ✓            | ✓            | **52.9**  | **8.6**  |
> |----------|-----------|--------------|--------------|-------|------|
> | ✗        | ✓         | ✓            | ✓            | 48.8  | 7.1  |
> | ✓        | ✗         | ✓            | ✓            | 49.7  | 7.8  |
> | ✓        | ✓         | ✗            | ✓            | 48.9  | 6.9  |
> | ✓        | ✓         | ✓            | ✗            | 49.5  | 7.5  |
> |----------|-----------|--------------|--------------|-------|------|
> | ✓        | ✗         | ✗            | ✗            | 48.1  | 6.3  |
> | ✗        | ✓         | ✗            | ✗            | 47.6  | 6.0  |
> | ✗        | ✗         | ✓            | ✗            | 48.5  | 6.2  |
> | ✗        | ✗         | ✗            | ✓            | 47.8  | 6.3  |
>
>
>
> **#3: It is also helpful to release the questions/answers generated on GSM8k and MATH for future comparisons and analysis.**
>
> We recognize the value of this for future comparisons and analysis, and will release both the generated dataset and the codebase used to create it.

---

> > ### Author Response · Authors · 2023-11-20
> > **Response to reviewer 1gMq (continued)**
> >
> > **#4: The paper only studies EAI on mathematical reasoning task although the proposed paradigm is quite simple and general. More thorough study on different tasks would better demonstrate the effectiveness of EAI in generating new data.**
> >
> > We agree with the reviewer that more thorough evaluation on a variety of benchmarks can make the paper more convincing. To this end, we conducted experiments on the code generation task. Results are shown in Table 5 in the paper and attached below as well. Evaluated with 3-shot MBPP test split and 0-shot HumanEval, EAI substantially outperforms all baselines and improve LLaMA2 and CodeLLaMA in their code generation performance, achieving 2.8% absolute increase in 3-shot MBPP and 1.9% increase in 0-shot HumanEval, confirming its effectiveness in exploring diverse supervision.
> >
> > |                    | LLaMA2 | LLaMA2+SFT | LLaMA2+RFT | LLaMA2+**EAI** |
> > |--------------------|-------:|-----------:|-----------:|-------------------:|
> > | **MBPP (%)**       |   26.1 |       38.5 |       41.8 | **44.6** _(+2.8)_  |
> > | **HumanEval**      |   11.9 |       13.2 |       13.9 | **16.2** _(+2.3)_  |
> >
> > |                    | CodeLLaMA | CodeLLaMA+SFT | CodeLLaMA+RFT | CodeLLaMA+**EAI** |
> > |--------------------|----------:|--------------:|--------------:|----------------------:|
> > | **MBPP (%)**       |      52.5 |          54.3 |          55.6 | **58.9** _(+3.3)_     |
> > | **HumanEval**      |      31.1 |          33.5 |          36.1 | **39.5** _(+3.4)_     |
> >
> >
> >
> > **#5: Can we introduce the actor-critic paradigm during inference, will this procedural inference help the reasoning on GSM8k?**
> >
> > This is a great idea: having an actor-critic during inference, where the actor generates diverse reasoning paths and the critic evaluates and guides these generations, could potentially improve decoding and reasoning. We are interested in this concept and plan to investigate it further in our follow-up work.
> >
> >
> > **#6: Other suggestions**
> >
> > Thanks for suggesting and we have incorporated them to improve the paper.
> >
> >
> > In conclusion, we believe that addressing these points substantially strengthens our submission. We hope that our response addresses your concerns. If our response addresses your concerns, please consider revising the score accordingly, otherwise, please let us know if anything further is required. We look forward to hearing from you.

---

### Official Review · Reviewer_uvsC · 2023-11-01

**Soundness:** 1 poor
**Presentation:** 3 good
**Contribution:** 1 poor
**Rating:** 3
**Confidence:** 3

**Summary:**

This work proposes self-guided exploration algorithms to teach LLMs to do complex tasks by having them generate data for them. The idea, in simple words, is that 1. Fine-tuning on small datasets of examples of that complex task improves the performance of the LLMs on that task, 2. LLMs are good at generating new data given few shot prompts, so 3. the authors propose that given some examples and guiding principles, LLMs can generate new training data for the complex task which it can be then fine-tuned on. However, the authors go a step beyond, and try to ensure that the novel generated data abides by some standards of correctness and diversity.

The authors pick two math benchmarks to evaluate this performance, GSM8K and MATH, which consist of math problems at different advancement level. The algorithm boils down to roughly the following: given an initial dataset of fine-tuning examples, and a set of "principles" (i.e. for math problems, rephrasing/restructuring the problem), the algorithm tries to generate a new example, and then based on an LLM critic's response on whether the new example is correct and diverse, it includes the new example in the dataset. The authors cite an RL-based unsupervised skill discovery algorithm as their inspiration.

Compared to the base Vicuna models, and also fine-tuned Vicuna models on the seed datasets, models trained on this augmented dataset perform better, respectively about ~5% on GSM8K dataset and ~3% on the MATH dataset.

The authors show some additional experiments, such as it helps to sample more data from the dataset while generating new points, and that their principle shows positive scaling with more model generation and human annotation. Moreover, more exploration principles result in a better downstream performance, which aligns with the intuitive understanding of this process.

**Strengths:**

1. The paper is presented well, including the initial idea of principle based exploration, and similarly, showing the prompts for actors, critics, and principles.
2. The problem proposed is interesting, we know that LLM performance in various field specific problems scale with available carefully annotated data, and getting human annotation for such data is difficult. If we could automate such generation that would be good.
3. Compared to RFT and SFT, the proposed EAI method has better performance in the mentioned benchmarks.

**Weaknesses:**

1. The paper only evaluates the proposed algorithm on a very narrow set of problems, namely only two benchmarks, and both relating to math problems. A more thorough evaluation on a variety of benchmarks covering different types of problems would be much more convincing re: the scalability of the method.
2. Moreover, using a math benchmark to evaluate this benchmark seems problematic since the LLM "critic" is also supposed to judge the correctness of the generated data point. However, with what we know about the hallucination problem in LLMs, it may not be a robust way to evaluate correctness.
3. The paper compares the numbers to very weak baselines like SFT/RFT + Vicuna while much stronger baselines like WizardMath and MAmmoTH are mentioned in the paper and Table 1. This shows that while this method is intellectually interesting, there are much better ways of generating a better finetuning dataset out there. No solid comparison with the stronger baselines beyond the throwaway numbers on Table 1 (a) makes it look suspicious and (b) makes scientific progress difficult, since future practitioners can't get solid insights to improve upon the proposed method without doing everything from scratch themselves. This is my primary complaint, and the major reason why I think the paper is unfit for publication in its current form by not contributing enough to our current state of knowledge.
4. The way of evaluating diversity in the algorithm 1 seems lacking; it is only checking local diversity and not global diversity. As a result, it is difficult to tell if the algorithm will scale or converge to some suboptimal local optima re: dataset creation.
5. There is no justification for picking the number 48K for number of generated datapoints. What happens when we keep increasing the number of new datapoints? Where does the limiting behavior occur?

Minor issues:
1. Figure 4 is wrong, it has two SFTs and green lines.
2. The rejection sampling based dataset generation method is not explained in enough detail, and thus it is hard to understand the primary baseline the authors compare against.
3. How is EAI generating anything without any "seed" data? (Figure 5). Similarly, how is this plot X axis going up to 8K when the human annotation only goes up to 7.5K data points (Table 1).
4. How is LLaMa SFT supervised by Human + LLaMa but Vicuna SFT supervised by Human only? What is the difference?

**Questions:**

Please see above in the Weakness section.

---

> ### Author Response · Authors · 2023-11-20
> **Response to review uvsC**
>
> Dear reviewer uvsC,
>
>
> Thank you so much for your review and for highlighting the idea of principle based exploration is well presented, and the research problem is interesting. We appreciate your feedback and suggestions, which we have incorporated into the revised version to improve it.
>
> Your main concern is comparing with stronger baselines. We will address your concern and answer your clarification questions below. Please let us know if our response satisfies you.
>
>
> **#1: Primary complaint: compare with much stronger baselines like WizardMath and MAmmoTH**
>
> We have conducted experiments comparing EAI with WizardMath and MAmmoTH. Previously, we did not directly compare them because their methods involve generating data from much more powerful models such as GPT-4 to improve LLaMA, while our experiments focus on generating data from LLaMA itself. Since we do not have access to GPT-4's weights, distilling GPT-4 is not transparent. Additionally, GPT-4 may change or discontinue service in the future, which could hinder reproducibility. Furthermore, the terms of service of OpenAI impose legal requirements on distilling their proprietary data. Therefore, we focused on enhancing LLaMA and Vicuna using the outputs generated by these models.
>
> That being said, we agree with the reviewer that a comparison with WizardMath and MAmmoTH can make the paper more convincing. We included experiments where we applied EAI to GPT-4 and then finetuned LLaMA with the resulting data, following the approaches of WizardMath and MAmmoTH. The results in Table 1 show that EAI performs substantially better than both approaches on both GSM8K and MATH, achieving new state-of-the-art performance and demonstrating EAI's effectiveness in generating diverse outputs.
>
>
>
> **#2: The paper only evaluates the proposed algorithm on a very narrow set of problems, namely only two benchmarks, and both relating to math problems. A more thorough evaluation on a variety of benchmarks covering different types of problems would be much more convincing re: the scalability of the method.**
>
> We focused on evaluating EAI on GSM8K and MATH benchmarks because these two are the main benchmarks used in our baselines publications.  We agree with the reviewer that more thorough evaluation on a variety of benchmarks can make the paper more convincing.
> To this end, we conducted experiments on the code generation task. Results are shown in Table 5 in the paper and attached below as well. Evaluated with 3-shot MBPP test split and 0-shot HumanEval, EAI substantially outperforms all baselines and improve LLaMA2 and CodeLLaMA in their code generation performance, achieving 2.8% absolute increase in 3-shot MBPP and 1.9% increase in 0-shot HumanEval, confirming its effectiveness in exploring diverse supervision.
>
> |                    | LLaMA2 | LLaMA2+SFT | LLaMA2+RFT | LLaMA2+**EAI** |
> |--------------------|-------:|-----------:|-----------:|-------------------:|
> | **MBPP (%)**       |   26.1 |       38.5 |       41.8 | **44.6** _(+2.8)_  |
> | **HumanEval**      |   11.9 |       13.2 |       13.9 | **16.2** _(+2.3)_  |
>
> |                    | CodeLLaMA | CodeLLaMA+SFT | CodeLLaMA+RFT | CodeLLaMA+**EAI** |
> |--------------------|----------:|--------------:|--------------:|----------------------:|
> | **MBPP (%)**       |      52.5 |          54.3 |          55.6 | **58.9** _(+3.3)_     |
> | **HumanEval**      |      31.1 |          33.5 |          36.1 | **39.5** _(+3.4)_     |
>
>
> **#3: Justification for picking the number 48K for number of generated data points*
>
> Regarding picking the number 48K as the number of generated data points, it is the maximum number of generated data points reported by RFT and we choose it to achieve a fair comparison and to reduce inference cost. We have made this more clear in revision. We have conducted more experiments to scale up 48K to 96K to study the scaling trend more rigorously.
>
> Regarding the trend of scaling, Figure 4 shows the performance of different methods along with the number of generated data points, EAI keeps outperforming RFT and scales well from 8K to 48K to 96K, while baselines saturate after 40-50K. This result confirms the effectiveness of EAI and we believe that further improvements can be done by doing large-scale exploration, which we leave for future work.
>
>
> **#4: How does EAI work without any “seed” data**
>
> For zero-shot exploration without human-provided “seed” data, we start by instructing Vicuna with instructions like `generating a high school level math question` and randomly sample 32 questions as the “seed” dataset. We will make this more clear.

---

> > ### Author Response · Authors · 2023-11-20
> > **Response to review uvsC (continued)**
> >
> > **#5: Figure 5 plot X axis shows 8K instead of 7.5K data points.**
> >
> > In Table 1, The 8K x-axis label in fact denotes the 7.5K data points available in GSM8K, we have updated the x-axis label in revision.
> >
> >
> > **#6: Other minor suggestions**
> >
> > We have included them in the revision, thank you for suggesting.
> >
> >
> > In conclusion, we believe that addressing these points substantially strengthens our submission. We hope that our response addresses your concerns. If our response addresses your concerns, please consider revising the score accordingly, otherwise, please let us know if anything further is required. We look forward to hearing from you.

---

### Official Review · Reviewer_td87 · 2023-11-01

**Soundness:** 2 fair
**Presentation:** 3 good
**Contribution:** 2 fair
**Rating:** 3
**Confidence:** 3

**Summary:**

This paper propose EAI, an iterative framework to query LLM to generate extra training data to better achieve generalization. The author takes inspiration from unsupervised RL and use prompting to achieve similar idea. The resulting algorithm improves over baseline on mathematical reasoning task.

**Strengths:**

The problem of having less human involvement in finetuning is an important topic. The link between unsupervised RL and this problem is very interesting. The proposed method seems to work on the experiment.

**Weaknesses:**

1. To start with, I think the method is still relying a lot on human insights especially on some exact ways of generating "diverse" methods. The argument that this is a general method is not well supported enough, and I would love to see more experiments on different kinds of benchmark.

2. The paper is missing ablation to see how exactly the critic help the results. In other words, it needs to compare with similar methods like self-instruct.

3. While the diversity experiment in Fig3, it would be more interesting in seeing more visualization. Since you already have the embeddings, maybe do a T-SNE plot or something to better prove the diversity

4. Finally, the method seems to have very weak link to unsupervised RL pretraininig, and in fact the model is not trained at all but simply prompted. And I don't see any thing related "learning skills".

**Questions:**

See above

---

> ### Author Response · Authors · 2023-11-20
> **Response to reviewer td87**
>
> Dear reviewer td87,
>
>
> Thank you so much for your review and insightful comments. We appreciate your feedback and suggestions, which we have incorporated into the revised version to improve it.
>
>
> **#1: The method is still relying on humans to provide exploration principles.**
>
> It is true that humans provide exploration principles to LLM to guide autonomous exploration. It is much easier to provide these principles than to manually curate data. We used a set of four exploration principles including paraphrasing questions, coming up with a new topic, coming up with a new scenario, and restructuring questions, which are simple to come up with and does not require domain expertise. In contrast, humans to curate the dataset is both time consuming and require domain expertise. Compared with human supervision, the EAI framework generates diverse AI supervision with minimal human input, is both scalable and effective.
> To make EAI even more scalable, we plan to investigate generating principles by LLM itself in future work.
>
>
> **#2: would love to see more experiments on different kinds of benchmarks**
>
> We focused on evaluating EAI on GSM8K and MATH benchmarks because these two are the main benchmarks used in our baselines publications.  We agree with the reviewer that more thorough evaluation on a variety of benchmarks can make the paper more convincing.
> To this end, we conducted experiments on the code generation task. Results are shown in Table 5 in the paper and attached below as well. Evaluated with 3-shot MBPP test split and 0-shot HumanEval, EAI substantially outperforms all baselines and improve LLaMA2 and CodeLLaMA in their code generation performance, achieving 2.8% absolute increase in 3-shot MBPP and 1.9% increase in 0-shot HumanEval, confirming its effectiveness in exploring diverse supervision.
>
> |                    | LLaMA2 | LLaMA2+SFT | LLaMA2+RFT | LLaMA2+**EAI** |
> |--------------------|-------:|-----------:|-----------:|-------------------:|
> | **MBPP (%)**       |   26.1 |       38.5 |       41.8 | **44.6** _(+2.8)_  |
> | **HumanEval**      |   11.9 |       13.2 |       13.9 | **16.2** _(+2.3)_  |
>
> |                    | CodeLLaMA | CodeLLaMA+SFT | CodeLLaMA+RFT | CodeLLaMA+**EAI** |
> |--------------------|----------:|--------------:|--------------:|----------------------:|
> | **MBPP (%)**       |      52.5 |          54.3 |          55.6 | **58.9** _(+3.3)_     |
> | **HumanEval**      |      31.1 |          33.5 |          36.1 | **39.5** _(+3.4)_     |
>
>
>
> **#3: Include an ablation study on critic**
>
> We recognize the importance of an ablation study to isolate the effect of the critic component in our framework. In the revised version, we include a detailed ablation study with its results shown in Table 4, comparing EAI with and without the critic component on GSM8K, MATH, MBPP, and HumanEval benchmarks.
>
> The results are also attached below for convenience. They show that EAI without the critic significantly outperforms the established baselines, demonstrating the effectiveness of principles guiding context exploration. Reintroducing the critic into the EAI framework substantially improves these results, achieving significantly better performance than Self-instruct, RFT, and EAI without the critic.
>
> A qualitative analysis provided in Appendix A reveals how the critic aids in guiding exploration. Both the quantitative and qualitative results demonstrate that the critic is crucial for achieving optimal exploration outcomes; removing the critic leads to substantially lower performance.
>
>
> |                   | LLaMA2 | Self-Instruct | RFT   | EAI w/o critic | **EAI** |
> |-------------------|--------|---------------|-------|---------------------|--------------|
> | GSM8K (%)         | 14.6   | 43.4          | 47.5  | 50.5                | **52.9**     |
> | MATH (%)          | 2.5    | 3.9           | 5.6   | 7.1                 | **8.6**      |
> | MBPP (%)          | 26.1   | 33.7          | 41.8  | 42.5                | **44.6**     |
> | HumanEval (%)     | 11.9   | 12.8          | 13.9  | 14.5                | **15.8**     |
>
>
>
>
> **#4: Include T-SNE results in addition to diversity gain.**
>
> We appreciate your suggestion to include more visualization for the diversity experiment. In response, we integrate T-SNE plots of the embeddings in Figure 3 of our revised manuscript. Compared with human curated GSM8K and RFT generated data points, EAI produces substantially more diverse data points,  providing more compelling evidence of its effectiveness.

---

> ### Author Response · Authors · 2023-11-20
> **Response to reviewer td87 (continued)**
>
> **#5: More discussions on connection to unsupervised RL pretraining and exploration**
>
> We will elaborate on the connection between our framework and unsupervised reinforcement learning (RL) in the revised paper. Although our method primarily uses prompting, it aligns with the principles of unsupervised RL by learning from the environment (in this case, the responses from the LLM) and adapting accordingly. Unsupervised RL like APT tries to visit states that are different from states sampled from replay buffer through reinforcement learning, EAI instead tries to generate data that are different from data sampled from replay buffer through principles guided in context exploration. We will also clarify how EAI contributes to the learning of unsupervised exploration over time through diverse in context examples, bridging the perceived gap.
>
>
> In conclusion, we believe that addressing these points substantially strengthens our submission. We hope that our response addresses your concerns. If our response addresses your concerns, please consider revising the score accordingly, otherwise, please let us know if anything further is required. We look forward to hearing from you.

---

### Official Review · Reviewer_7sTU · 2023-11-03

**Soundness:** 4 excellent
**Presentation:** 3 good
**Contribution:** 3 good
**Rating:** 6
**Confidence:** 4

**Summary:**

This paper presents Exploratory AI (EAI) to generate diverse instruction-tuning data that can further improve large language models (LLMs). EAI leverages unsupervised RL pre-training to explore within the natural language space. Experiments show that EAI can significantly boost the performance on complex reasoning datasets.

**Strengths:**

+ This paper is well-written and easy to follow.
+ Instruction-tuning data is crucial to LLMs. Automatically generating them for training is a practical direction to avoid elaborate human annotations.
+  The proposed EAI brings notable improvements (Table 1), which demonstrates its effectiveness.

**Weaknesses:**

+ What is the training efficiency of EAI? From my best understanding, it will take lots of overhead for this RL pre-training.
+ From Table 2, it seems that a larger replay buffer can achieve more improvements. What if we use an even larger one (e.g., 12 or 16)? Will the performance keep increasing or converge?
+ There should be a detailed analysis of the quality/diversity of the generated content (not just performance-wise evaluation). For example, a human evaluation to investigate them.
+ Some qualitative results of the generated content should be presented, including both successful and failed cases.
+ There is a critic to evaluate the generated content. However, since the critic is also the LLM as the actor, how if both have the same blind spot and derive a wrong evaluation? This may further hurt the fine-tuning.
+ The format seems not to be ICLR. Not sure if we should desk reject this draft.

**Questions:**

Please see the weakness

---

> ### Author Response · Authors · 2023-11-20
> **Response to reviewer 7sTU**
>
> Dear reviewer 7sTU,
>
>
> Thank you so much for your positive review and insightful comments. We appreciate your feedback and suggestions, which we have incorporated into the revised version to improve it.
>
>
> **#1: From Table 2, it seems that a larger replay buffer can achieve more improvements. What if we use an even larger one (e.g., 12 or 16)? Will the performance keep increasing or converge?**
>
> We conducted experiments to investigate this. We scale up the number of samples from replay buffer from 8  to 12 and 16, and evaluate the performance. The following table presents the results of an experiment examining the impact of varying the number of samples on GSM8K and MATH.
> As the number keeps increasing, the performance on both datasets keep increasing, highlighting the significance of having larger input size for exploration.
> LLM with larger context window size can fit in more samples, and potentially lead to more effective exploration.
>
> |        |  0  |  1  |  4  |  8  | 12  | **16** |
> |--------|-----|-----|-----|-----|-----|--------|
> | GSM8K  | 50.1| 50.8| 51.9| 52.9| 53.5| **54.2**|
> | MATH   | 6.6 | 7.1 | 7.5 | 9.6 |10.8 | **11.6**|
>
>
>
> **#2: What is the training efficiency of EAI?**
>
> EAI is compute intensive due to it involves sampling from large language models. Using 8x A100, generating 48K samples with EAI takes about 12 hours. As inference approaches keep advancing, the training efficiency of EAI will keep increasing.
>
>
> **#3: Include a detailed analysis of the quality/diversity of the generated content.**
>
> We appreciate your suggestion to include more visualization for the diversity experiment. In response, we integrate T-SNE plots of the embeddings in Figure 3 of our revised manuscript. Compared with human curated GSM8K and RFT generated data points, EAI produces substantially more diverse data points,  providing clearer and more compelling evidence of its effectiveness.
>
>
> **#4: Include qualitative results of the generated content.**
>
> We appreciate your suggestion to include qualitative analysis. A qualitative analysis provided in Appendix A, which  reveals how the EAI accomplishes exploration. Actor follows principles to generate diverse samples and the critic provides feedback to aid the exploration.
>
> **#5: Potential issues of critic and actor being the same LLM.**
>
> Across all four benchmarks we evaluated, we did not observe any degeneration due to exploration. Since the actor is guided by a set of exploration principles, such as 'restructuring questions', its outputs are steered towards high-quality data. Moreover, the critic's method of evaluating the actor’s outputs, akin to widely used self-reflection techniques, further enhances the quality of exploration. Furthermore, the benefits of diverse samples may outweigh the drawbacks of a potentially incorrect subset of samples.
>
>
> **#6: Other suggestions.**
>
> Thanks for suggesting and we have incorporated them to improve the paper.
>
>
> In conclusion, we believe that addressing these points substantially strengthens our submission. We hope that our response addresses your concerns. If our response addresses your concerns, please consider revising the score accordingly, otherwise, please let us know if anything further is required. We look forward to hearing from you.

---

> > ### Comment · Reviewer_7sTU · 2023-11-20
> >
> > Thanks for the response, which mostly resolves my concerns. As we also agree with issues raised by other reviewers, I keep my score as borderline accept.

---

### Meta-Review · Area_Chair_S5q3 · 2023-12-07

**Metareview:**

This paper studies a self-supervised approach for generating synthetic training data by prompting an LLM as both an "actor" that generates new data and a "critic" that evaluates the novelty and correctness of the generated data. The experiments focus on GSM8K and MATH, where the method achieves improvements compared to SFT and RFT baselines. The idea itself is simple and compelling.

However, reviewers point out a few important weaknesses in this work:
- The authors should evaluate their method on a broader set of tasks, beyond simply math and coding, as the method is presented as a general approach to generating synthetic data.
- Some reviewers point out that it is somewhat confusing that the approach is motivated from the lens of an actor-critic RL algorithm, when the relationship to AC algorithms is at best a loose analogy. This presentation introduces a somewhat spurious relationship to RL, when there is not any deep connection between AC algorithms and EAI articulated in this work.
- Not enough ablations are conducted on the design of the actor and critic prompts themselves. There is no ablation testing the effect of alternative prompts.
- There is little analysis to support the idea that the generated data is more diverse. In fact, the measure of diversity in this work is not even defined.
- The critic ablation shows little change in results, indicating one core half of the method may not even be necessary. These small differences in performances are also hard to interpret as there are no error bars presented in the reported results.

**Justification For Why Not Higher Score:**

The paper is recommended for rejection due to the issues summarized in the meta-review and highlighted by the reviewers.

**Justification For Why Not Lower Score:**

N/A

---

### Decision · Program_Chairs · 2024-01-16

Reject